# Low Tenacity of *Toxoplasma gondii* Tachyzoites In Vitro

**DOI:** 10.3390/microorganisms13071517

**Published:** 2025-06-29

**Authors:** Thomas Grochow, Mirjam Kalusa, Pauline Tonndorf-Martini, Nicole Röhrmann, Simone A. Fietz, Lea-Christina Murnik

**Affiliations:** 1Institute of Veterinary Anatomy, Histology and Embryology, Faculty of Veterinary Medicine, Leipzig University, 04103 Leipzig, Germany; mirjam.kalusa@vetmed.uni-leipzig.de (M.K.); pauline.tonndorf@gmail.com (P.T.-M.); nicole.roehrmann@vetmed.uni-leipzig.de (N.R.); simone.fietz@vetmed.uni-leipzig.de (S.A.F.); 2Institute of Parasitology, Faculty of Veterinary Medicine, Leipzig University, 04103 Leipzig, Germany; l.murnik@gmx.de

**Keywords:** *Toxoplasma gondii*, in vitro, cell culture, tenacity, tachyzoites, disinfection, chemical treatment, heat inactivation

## Abstract

*Toxoplasma gondii*, an obligate intracellular protozoan, poses significant risks to public health due to its widespread distribution and potential for severe congenital and neurological complications. The fast-replicating tachyzoite stage is crucial for acute infection and laboratory studies, yet effective inactivation methods remain inadequately explored. This study evaluates various chemical and physical approaches to inactivate *T. gondii* tachyzoites in vitro. Using a combination of GFP fluorescence and viability assays, we demonstrated the complete inactivation of tachyzoites with ethanol (≥30%), hydrogen peroxide (≥3%), o-hydroxydiphenyl fatty acid eutectic with peracetic acid (≥1%), and heat treatment at 60 °C for 30 min. Our findings highlight the importance of concentration, solvent choice, and exposure time in disinfection efficacy, providing a framework for improving laboratory safety protocols. These results contribute to the refinement of inactivation strategies, supporting safer handling and research on *T. gondii* in vitro while reducing reliance on animal models.

## 1. Introduction

Toxoplasmosis, a disease caused by the obligate intracellular parasite *Toxoplasma* (*T*.) *gondii*, is a widespread zoonosis with significant implications for both human and animal health [1,2]. The parasite infects a wide range of warm-blooded animals, including humans, serving as intermediate hosts, while felids act as definitive hosts where sexual reproduction occurs in the intestinal epithelium, leading to the shedding of environmentally resistant oocysts [3,4,5]. These oocysts have the ability to contaminate soil, water, and food sources, maintaining the cycle of infection [1,6]. Globally, *T*. *gondii* seroprevalence affects approximately one-third of the human population [1,7,8], with infections generally being asymptomatic in immunocompetent individuals but capable of causing severe consequences in cases of congenital transmission or in immunosuppression [7].

The parasite transitions between three infectious stages: tachyzoites, bradyzoites, and oocysts. Infection is transmitted through the ingestion of sporulated oocysts or tissue cysts present in undercooked or raw meat. Following ingestion, the parasite releases sporocysts that invade host cells, where they transform into tachyzoites, the rapidly replicating form of the parasite. These tachyzoites then disseminate through the host via the lympho-hematogenous route, targeting tissues such as the brain, heart, and muscles. In response to host immunity, tachyzoites differentiate into bradyzoites, forming tissue cysts that persist for the host’s lifetime [1,5]. Chronic *T*. *gondii* infection in humans and other mammals may be associated with neurological complications, including encephalitis and behavioral changes, particularly in immunosuppressed individuals [9,10]. Furthermore, congenital *T*. *gondii* infection may pose severe risks, as the parasite can cross the placental barrier during maternal parasitemia, resulting in the infection of the developing fetus [1,5]. In humans, the consequences of congenital toxoplasmosis vary depending on the timing of infection during gestation, ranging from abortion in early pregnancy to severe neurodevelopmental impairments, such as microcephaly and intellectual disability [7].

In recent years, the in vitro use of *Toxoplasma* has gained increasing attention in research, particularly in the context of the 3R principle (refine, replace, reduce). Advances in cell culture techniques have enabled the reliable replication of *Toxoplasma* by producing tachyzoites and initiating cyst formation under controlled laboratory conditions [11,12]. These in vitro methods offer valuable opportunities to study the biology and pathogenicity of the parasite, while reducing reliance on animal experiments. However, ensuring laboratory safety when working with *Toxoplasma* remains a critical issue, highlighting the importance of effective inactivation methods for its various developmental stages.

Effective disinfection methods are crucial for controlling the transmission of *T*. *gondii* and creating a safe environment when working with this parasite under research settings. Disinfection options for *T*. *gondii* encompass various methods, targeting both oocysts and tissue cysts. Chemical disinfectants (for example alcohol, acids, sodium chloride, and formaldehyde) have been evaluated for their efficacy in inactivating oocysts. However, the results of these studies have been inconsistent, with traditional chemical treatments demonstrating only limited success [13]. In Germany, two large disinfectant groups—oxygen scavengers and cresols—have been approved (according to the DVG-Desinfektionsmittelliste für den Tierhaltungsbereich [14]) for the disinfection of parasitic protozoa. In contrast, physical methods, such as high-pressure processing and ionizing radiation, have exhibited superior efficacy [13,15]. For tissue cysts, thermal treatments such as heating and freezing have proven effective, with appropriate cooking (60–70 °C) significantly reducing cyst viability [15]. While pharmaceutical approaches, including the use of atovaquone and azithromycin, have demonstrated some efficacy in reducing tissue cysts in vitro, there are currently no approved drugs capable of eradicating them completely [16]. To the best of the authors’ knowledge, there are no reported disinfection methods for the fast-replicating stage of *Toxoplasma*, the tachyzoites, even though the disinfection of these forms is highly relevant when working with *T*. *gondii* in vitro.

The objective of the present study was to evaluate the efficacy of various disinfection methods for *T*. *gondii* tachyzoites in cell cultures. The findings of this study offer valuable preliminary insights into the effectiveness of these methods, providing a foundation for future investigations aimed at optimizing *T*. *gondii* disinfection strategies.

## 2. Materials and Methods

### 2.1. Tachyzoites

MARC-145 monolayers (kindly provided by Gereon Schares, National Reference Laboratory for Toxoplasmosis, Institute of Epidemiology, Friedrich-Loeffler-Institut, Federal Research Institute for Animal Health, Greifswald-Insel Riems, Germany, ATCC^®^ CRL-12231) were utilized as the infection model. MARC-145 cells were seeded (2 × 10^5^ cells/well) in 100 mm cell culture dishes containing Dulbecco’s Modified Eagle’s Medium (DMEM, Gibco^TM^, ThermoFisher Scientific, Waltham, MA, USA) supplemented with 10% fetal bovine serum (FBS PAN Biotech^TM^, Aidenbach, Germany), 100 IU penicillin, 100 μg/mL streptomycin, (Pen Strep, Gibco^TM^, ThermoFisher Scientific, Waltham, MA, USA), and 2.5 μg/mL amphotericin B (Amphotericin B (PAN Biotech^TM^, Aidenbach, Germany). The cultures were incubated at 37 °C in a humidified atmosphere with 5% CO_2_ until they reached approximately 80% confluency. Confluent MARC-145 monolayers were inoculated with transgenic autofluorescent *T*. *gondii* RH-GFP tachyzoites (type I strain, kindly provided by Professor Dominique Soldati-Favre, University of Geneva Medical School, Geneva, Switzerland). From the time of inoculation, the FBS concentration in the medium was reduced to 2%. At 24 h post-infection, the infected cells were washed three times with sterile phosphate-buffered saline (PBS, pH 7.2), and fresh medium was added. After an additional 96 h, the medium containing free tachyzoites was removed, and the cells were washed three times with PBS. Centrifugation steps were performed at 400× *g* for 5 min. Tachyzoite counts were determined using a Neubauer improved hemocytometer.

### 2.2. Tachyzoite Treatment

For each experimental group, 5 × 10^5^ tachyzoites in 500 µL incubation medium were subjected to various chemical or heat treatments. Samples undergoing chemical treatment were incubated under the same culture conditions as described above. Heat treatments were performed using a Thermomixer Comfort (350 Hz, Eppendorf SE, Hamburg, Germany). The heat treatments were terminated by rapidly cooling the Thermomixer to 37 °C immediately following the incubation period. All experiments were performed a total of three times.

After 25 or 5 min of chemical treatment, the samples were centrifuged at 400× *g* for 5 min and washed three times with PBS, resulting in total treatment durations of 30 m and 10 m, respectively. Detailed treatment protocols are provided in Table 1. All experiments were performed a total of three times.

The reagents were purchased from the following manufacturers: ethanol and H_2_O_2_ (T171.8 and 9681.4, Carl Roth GmbH & Co. KG, Karlsruhe, Germany), p-chloro-m-cresol (Neopredisan 135-1, Menno-Chemie Vertrieb GmbH, Norderstedt, Germany), o-hydroxydiphenyl fatty acid eutectic and in component + peracetic acid (Ascarosteril^®^ AB, Kesla Pharma Wolfen GmbH, Wolfen, Germany; the two components were mixed according to the manufacturer’s instructions).

### 2.3. Live/Dead Staining

Live/dead staining was conducted to evaluate tachyzoite viability. For this purpose, 1 × 10^5^ tachyzoites were suspended in 100 µL PBS and stained with 4′,6-diamidino-2-phenylindole (DAPI; 1:500, Sigma-Aldrich Corp., Taufkirchen, Germany). Stained samples were immediately analyzed under a fluorescence microscope (400× magnification, Zeiss Axiophot, Carl Zeiss Microscopy Deutschland GmbH, Oberkochen, Germany).

For each sample, 100 tachyzoites were randomly selected and evaluated. Viable cells with intact membranes exhibited minimal DAPI uptake, resulting in weak DNA staining. In contrast, non-viable cells, characterized by compromised membranes, showed intense DAPI staining (Figure 1a,b).

In this experiment, transgenic GFP-labeled tachyzoites were utilized. In viable tachyzoites, GFP was evenly distributed throughout the cytoplasm. Conversely, non-viable cells displayed either focal aggregation or a complete loss of GFP signal (Figure 1a,b).

### 2.4. Ability to Reproduce

To evaluate whether the treated tachyzoites retained the ability to reproduce, they were re-cultured on a cell monolayer. MARC-145 cells were seeded in a 96-well plate (655180, Greiner Bio-One International GmbH, Kremsmünster, Austria) and grown to approximately 80% confluency under the previously described culture conditions. Treated tachyzoites were inoculated onto the monolayers at a density of 1.28 × 10^5^ tachyzoites per well, corresponding to a multiplicity of infection (MOI) of four.

After one week of incubation, the proliferation of the GFP-labeled tachyzoites was evaluated using confocal laser scanning microscopy (cLSM) with a Leica TCS SP8 system (Leica Microsystems, Mannheim, Germany) and Leica Application Suite X (LAS-X, version 3.5.7) software. Images were acquired using the following microscope settings: bidirectional scanning, scan speed of 600 Hz, pinhole size of 1 Airy unit, and 3 × line averaging. Sequential channel scanning was employed to minimize crosstalk between dyes. GFP fluorescence was excited at 488 nm (1% laser intensity, PMT detection range: 493–600 nm). Tile scans of 3270 × 3270 μm^2^ were acquired centrally in the wells using a 10×/0.40 dry objective.

Deconvolution of cLSM images was performed using Huygens Professional software (version 20.10, SVI, Hilversum, The Netherlands). Parameters were set according to image metadata, with a theoretical point-spread function applied. Batch processing employed the Classic Maximum Likelihood Estimation algorithm with the following settings: maximum iterations = 40, quality change threshold = 0.01, signal-to-noise ratio = 7, and background = 0.

Image analysis and quantification of GFP-positive areas were conducted using Imaris software (version 10.1, Oxford Instruments, Abingdon, UK). Surface rendering was performed with the following settings: surface detail = 5 μm, manual threshold value = 10, and quality filter > 1000. The total area of GFP-positive signals was quantified for each tile scan.

All experiments were performed a total of three times.

### 2.5. Statistical Analysis

Statistical analyses were performed using a GraphPad Prism (version 6.0f, GraphPad Software Inc., San Diego, CA, USA). An ordinary one-way ANOVA, followed by Dunnett’s post hoc test, was conducted. *p*-values < 0.05 were considered statistically significant. Particular attention was given to identifying values that did not differ significantly from the positive control. All data are presented as mean ± standard error of the mean (SEM).

## 3. Results

### 3.1. Evaluation of Inactivation Methods on Tachyzoite Viability and Proliferation

Different approaches were tested to evaluate the efficacy of different inactivation methods on transgenic GFP-labeled tachyzoites (*T*. *gondii* strain RH-GFP). A total of 5 × 10^5^ tachyzoites were exposed to stressors for 30 or 10 min (Table 1). After treatment, 1 × 10^5^ tachyzoites were stained with DAPI to assess viability. DAPI, a proven live/dead dye, showed low uptake in viable cells with intact membranes. In viable cells, GFP fluorescence remained homogeneously distributed throughout the cytoplasm (Figure 1a). Conversely, in non-viable tachyzoites with compromised membrane integrity, DAPI influx occurred, resulting in DNA intercalation, particularly in the nucleus (Figure 1b).

Treated tachyzoites were also inoculated onto 80% confluent MARC-145 monolayers at a density of 1.28 × 10^5^ tachyzoites per well (MOI = 4). After 1 week of incubation, the proliferation of GFP-labeled tachyzoites was evaluated using cLSM and quantified with Imaris software.

### 3.2. Impact of Treatments on Tachyzoite Survival

#### 3.2.1. Controls

PBS served as a negative control, yielding a survival rate of 90.67 ± 1.76%. The positive control, 4% paraformaldehyde in PBS, is a commonly used substance in laboratories to inactivate and fix tachyzoites [17,18]. It resulted in the complete inactivation of the parasite with a survival rate of 0%.

#### 3.2.2. Ethanol Treatments

Ethanol diluted in double-distilled water was first assessed. Across all concentrations, including 0% (pure distilled water) up to 70%, the survival rate was 0%, showing no significant difference from the positive control (*p* > 0.9999, Figure 1c). When ethanol was diluted in PBS, 10% ethanol exhibited low killing efficacy, with survival rates of 69.33 ± 2.67% after 30-min incubation and 77.33 ± 1.76% after 10-min incubation. However, concentrations exceeding 30% resulted in complete parasite elimination, rendering survival rates statistically indistinguishable from those of the positive control (*p* > 0.9999, Figure 1d).

#### 3.2.3. Hydrogen Peroxide Treatments

Hydrogen peroxide (H_2_O_2_) diluted in PBS achieved complete tachyzoite inactivation at concentrations of 3% or higher following 10-min incubation. At 0.3%, a considerable proportion of tachyzoites survived, with survival rates of 30.67 ± 1.76% after 30-min incubation and 55.33 ± 2.90% after 10-min incubation (Figure 1e).

##### 3.2.4. p-chloro-m-cresol Treatments

Treatment with p-chloro-m-cresol resulted in reductions in viability, but did not achieve full inactivation at any tested concentration or duration. The most effective condition, 4% for 30 min, reduced viability to 46.67 ± 1.76%, indicating residual survival (Figure 1f).

##### 3.2.5. o-Hydroxydiphenyl Fatty Acid Eutectic + Peracetic Acid Treatments

All tested concentrations, including 1 to 4.5%, and incubation times of o-hydroxydiphenyl fatty acid eutectic combined with peracetic acid resulted in complete tachyzoite inactivation, with survival rates not significantly different from those of the positive control (*p* > 0.9999, Figure 1g).

#### 3.2.6. Heat Treatments

Heat treatment was also investigated as a physical inactivation method. Incubation at 40 °C for 30 or 10 min and at 50 °C for 10 min resulted in survival rates of approximately 89%. At 50 °C for 30 min, a reduction in viability was observed, with survival rates decreasing to 42.67 ± 1.76%. However, complete inactivation required 60 °C for 30 min, where survival rates dropped to 0%. Notably, a minor proportion (3.33 ± 0.67%) survived after 10 min at 60 °C (Figure 1h).

### 3.3. Impact of Treatments on Tachyzoite Proliferation

#### 3.3.1. Controls

Mock-treated tachyzoites (PBS) exhibited abundant GFP fluorescence, indicative of successful proliferation (Figure 2a). In contrast, tachyzoites treated with the positive control (4% paraformaldehyde) showed no signs of proliferation, as evidenced by the absence of GFP-positive signals (Figure 2b).

#### 3.3.2. Ethanol Treatments

Ethanol diluted in double-distilled water was first assessed. Across all concentrations, from 0% (pure distilled water) up to 70%, no GFP fluorescence was detected, indicating no growth. These results were not significantly different from the positive control (*p* > 0.9999, Figure 2c). When ethanol was diluted in PBS, all concentrations from 30% resulted in no GFP fluorescence and no growth, rendering them not significantly different from the positive control (*p* > 0.9999). The exception was the samples treated with 10% ethanol, which exhibited residual GFP-positive areas of 2.26 × 10^5^ ± 3.74 × 10^4^ μm^2^ (30 min incubation) and 1.07 × 10^6^ ± 1.71 × 10^5^ μm^2^ (10 min incubation), indicating significant growth compared with the positive control (*p* [10%, 10 min] < 0.0001, Figure 2d).

#### 3.3.3. Hydrogen Peroxide Treatments

Hydrogen peroxide (H_2_O_2_) diluted in PBS achieved complete tachyzoite inactivation at concentrations of 3% or higher, as indicated by the absence of GFP fluorescence. These results were not significantly different from the positive control (*p* > 0.9999). However, the sample treated with 0.3% H_2_O_2_ for 10 min showed residual GFP-positive areas of 5.13 × 10^3^ ± 5.13 × 10^3^ μm^2^, indicating some growth but no significant difference compared to the positive control (*p* > 0.9999, Figure 2e).

#### 3.3.4. p-chloro-m-cresol Treatments

Treatment with p-chloro-m-cresol resulted in substantial GFP fluorescence at all tested concentrations and incubation times, with GFP-positive areas ranging from 2.74 × 10^5^ ± 2.28 × 10^4^ μm^2^ to 4.57 × 10^4^ ± 2.31 × 10^3^ μm^2^. These values showed growth, even if they did not differ significantly compared to the positive control (Figure 2f).

#### 3.3.5. o-Hydroxydiphenyl Fatty Acid Eutectic + Peracetic Acid Treatments

All tested concentrations and incubation times of o-hydroxydiphenyl fatty acid eutectic combined with peracetic acid resulted in complete inactivation, as no GFP fluorescence or growth was detected. These results were not significantly different from the positive control (*p* > 0.9999, Figure 2g).

#### 3.3.6. Heat Treatments

Heat treatment was evaluated as a physical inactivation method. Most conditions, including incubation at 40 °C and 50 °C for 30 or 10 min and 60 °C for 10 min, showed GFP fluorescence and growth, with GFP-positive areas ranging from 2.98 × 10^6^ ± 3.34 × 10^5^ μm^2^ to 6.20 × 10^3^ ± 1.94 × 10^3^ μm^2^. These results indicated significant differences compared with the positive control (*p* [40 °C, 30 and 10 min] < 0.0001). However, complete inactivation was achieved only at 60 °C for 30 min, where no GFP fluorescence or growth was detected, making these results not significantly different from the positive control (*p* > 0.9999, Figure 2h).

## 4. Discussion

The effective inactivation of *T*. *gondii* tachyzoites is critical for ensuring laboratory safety, particularly in the context of increasing efforts to reduce animal experimentation by utilizing in vitro systems. Our findings demonstrate that certain chemical and physical methods are capable of reliably inactivating tachyzoites, thereby offering practical options for researchers handling this infectious stage.

In this study, various concentrations of ethanol diluted in double-distilled water were initially tested for their efficacy against *T*. *gondii* tachyzoites. Surprisingly, complete inactivation was observed across all tested dilutions, including 0% ethanol (pure double-distilled water). This finding strongly suggests that osmotic stress induced by the hypotonic environment caused lysis of the tachyzoites, as similarly reported by Räisänen et al. [19], who reported that tachyzoites extracted from ascitic fluid are unable to survive in highly hypotonic solutions. These results underscore the critical influence of solvent composition on experimental outcomes. To eliminate the confounding effect of osmotic lysis, subsequent experiments were performed using PBS as the solvent, providing isotonic conditions. Under these conditions, a concentration-dependent effect of ethanol became apparent: ethanol concentrations of 30% or higher reliably achieved complete inactivation of the tachyzoites after 10 min of exposure. In contrast, lower concentrations, such as 10% ethanol, failed to ensure complete inactivation, likely due to insufficient membrane-disruptive properties. The results of this study show that ethanol-based disinfectants achieve reliable inactivation of *T*. *gondii* tachyzoites when used at sufficient concentrations under isotonic conditions. This finding is consistent with previous reports demonstrating that alcohols can effectively impair the membrane structures of other protozoan parasites, such as *Trypanosoma brucei* and *Leishmania major* [20]. In contrast to oocysts, which are characterized by a robust wall structure, tachyzoites exhibit a comparatively fragile cell membrane, which likely contributes to their higher susceptibility to chemical disinfection [5]. Traditional chemical disinfectants have generally been found to be insufficient for the inactivation of oocysts [13], underlining the necessity of stage-specific disinfection strategies when working with *T*. *gondii*.

Hydrogen peroxide (H_2_O_2_) demonstrated high efficacy in this study, with complete tachyzoite inactivation achieved after 10 min of exposure. This observation aligns with previous reports that hydrogen peroxide can effectively oxidize cellular components and impair membrane integrity in *T*. *gondii* tachyzoites and other protozoa, such as *Cryptosporidium parvum* and *Ichthyophthirius multifiliis* [21,22,23]. Its strong oxidizing properties likely contribute to the disruption of vital cellular structures in tachyzoites, making it a potent option for laboratory disinfection. However, it is important to note that while H_2_O_2_ is effective against tachyzoites, prior studies on *Eimeria acervulina*—another apicomplexian parasite—have shown that it is significantly less effective against the resistant oocyst stage [24].

In contrast, p-chloro-m-cresol, a cresolic disinfectant approved in Germany for the inactivation of protozoa in veterinary settings [14], showed only limited efficacy against *T*. *gondii* tachyzoites under the conditions tested. Despite its documented activity against more resistant stages such as oocysts, the incomplete inactivation observed in this study suggests that p-chloro-m-cresol may not be fully reliable when tachyzoites are the predominant stage of concern. The manufacturer’s recommended contact time of 120 min was not explored due to practical constraints, resulting in the underutilization of the full potential of this agent. This finding highlights the necessity of validating disinfection methods for specific parasite stages rather than assuming cross-stage efficacy.

The combination of o-hydroxydiphenyl fatty acid eutectic and peracetic acid was consistently effective, achieving complete inactivation across all tested concentrations and incubation times, thereby highlighting the synergistic action of these compounds and supporting their use as a robust disinfection method. Given its rapid action and broad-spectrum activity, the combination represents a highly effective option for inactivating tachyzoites under laboratory conditions. However, its application must be handled carefully due to its corrosive nature and potential health risks for laboratory personnel.

Thermal treatment also proved effective, with heating leading to reliable inactivation of tachyzoites. This is in line with previous findings, in which heating to 55–82 °C significantly reduced the viability of tissue cysts [15]. Given the delicate structure and high metabolic activity of tachyzoites, their sensitivity to heat stress appears biologically plausible and confirms the efficacy of heat-based disinfection methods for the tachyzoite stage. The survival rate of tachyzoites heated at 60 °C for 10 min was remarkably low (3.33%). However, varying data on the inactivation of tissue cysts can be found in the literature, with reported thresholds ranging from 58 °C for 9.5 min [25] to 60 °C for 10 min [26], or at least 67 °C for 10 min [27]. These discrepancies may be attributed to differences in experimental setups, such as tissue size or sample conditions. Notably, 60 °C for 10 min appears to lie precisely at the threshold for effective inactivation. Therefore, to ensure reliable and complete killing of tachyzoites, it is strongly recommended to extend the incubation time to 30 min.

This study provides a foundation for the safe laboratory handling of *T*. *gondii* tachyzoites by validating effective disinfection methods. However, this study is not without limitations. Notably, it excluded extended contact times for p-chloro-m-cresol and lacked direct comparisons with other protozoan stages. Future research should explore the cross-stage applicability of these methods and assess their long-term safety and environmental impacts.

## 5. Conclusions

In summary, this study successfully validated ethanol (≥30%), distilled water, hydrogen peroxide (≥3%), o-hydroxydiphenyl fatty acid eutectic with peracetic acid, and heat treatments (60 °C for 30 min) as reliable inactivation strategies for *T*. *gondii* tachyzoites, contributing to the advancement of safe laboratory practices in parasitology.

## Figures and Tables

**Figure 1 microorganisms-13-01517-f001:**
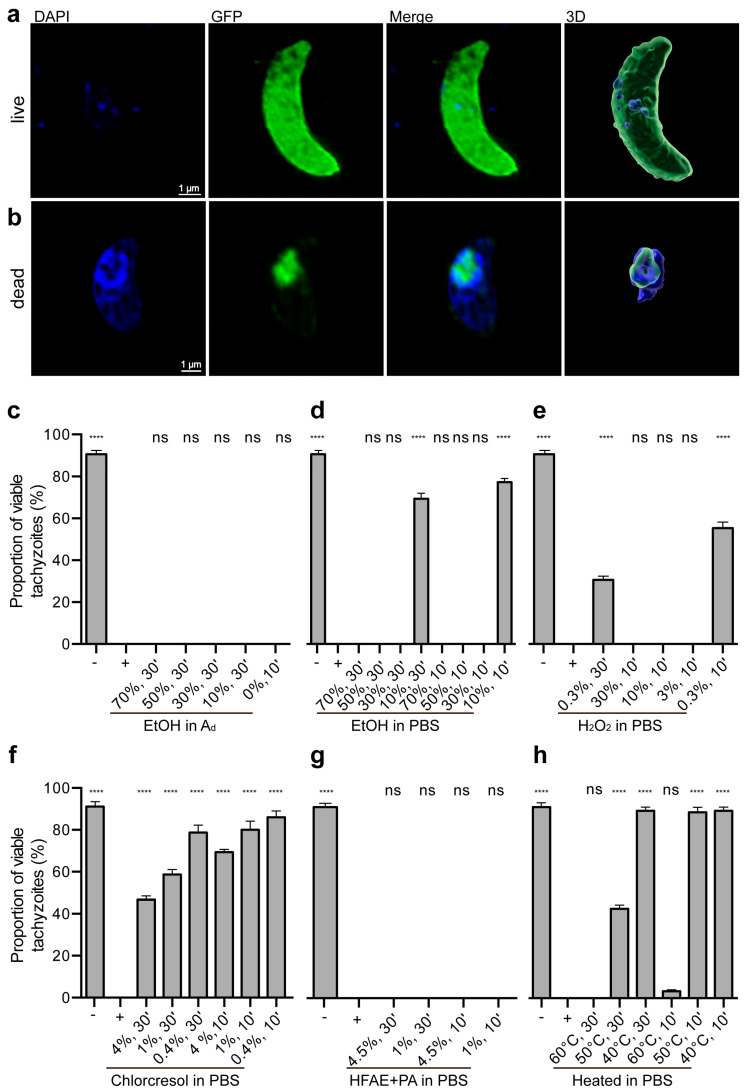
Survival of *T. gondii* tachyzoites after treatment. After treatment, DAPI live/dead staining was performed, and tachyzoites were immediately analyzed under a fluorescence microscope (**a**). (**b**) Maximum intensity projections of 17 and 25 optical sections, respectively, showing DAPI staining (blue) and GFP fluorescence for transgenic RH-GFP tachyzoites (green). For visualization, images were adjusted using Photoshop. Three-dimensional renderings were created by surface rendering with Imaris for visualization purposes only. (**c**–**h**) Proportion of live tachyzoites after treatment, determined via DAPI staining. The control data presented here are identical to those shown in (**c**–**h**), m. EtOH in Ad: ethanol diluted in double-distilled water; EtOH in PBS: ethanol diluted in PBS; H_2_O_2_ in PBS: hydrogen peroxide diluted in PBS; chlorcresol in PBS: p-chloro-m-cresol diluted in PBS; HFAE + PA in PBS: o-hydroxydiphenyl fatty acid eutectic component + peracetic acid diluted in PBS; ns: no significant difference compared to the positive control. All other groups showed significant differences (**** *p* < 0.0001) compared with the positive control. Individual *p*-values are provided in Appendix A. Error bar: standard error of the mean.

**Figure 2 microorganisms-13-01517-f002:**
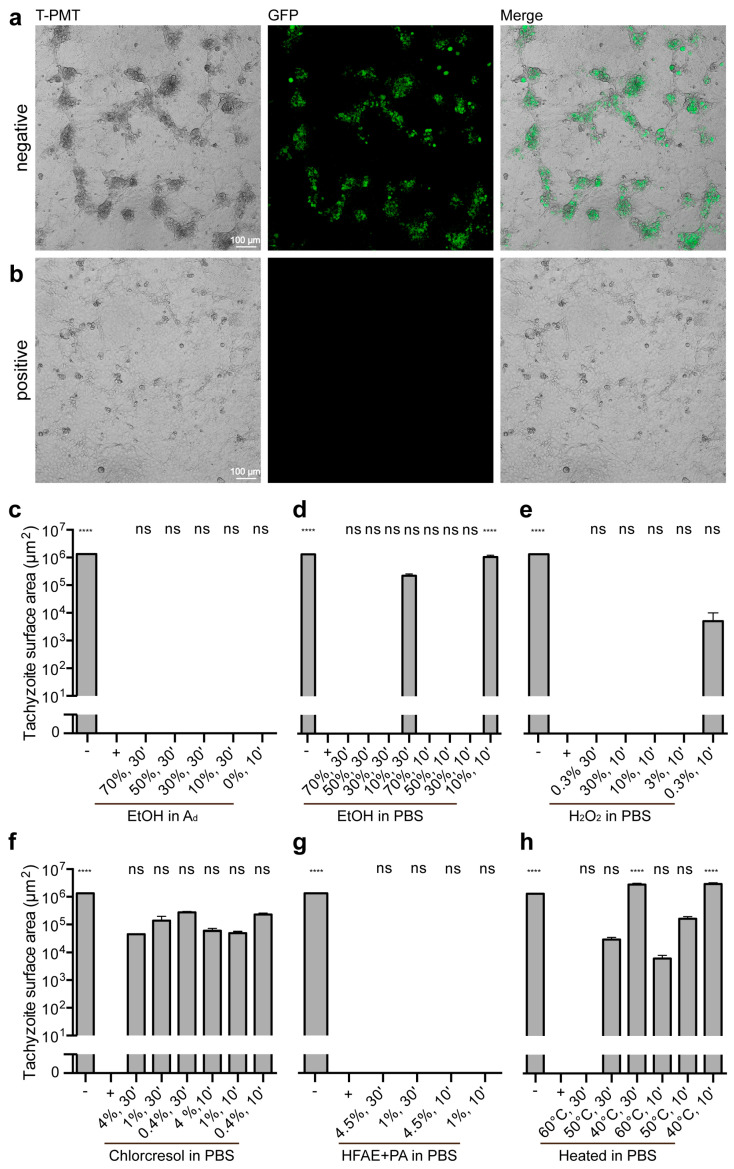
Tachyzoite growth after treatment. (**a**,**b**) Selected regions of analyzed confocal images showing immunofluorescence for transgenic RH-GFP tachyzoites (green), seeded on Marc 145-cells. For visualization purposes, T-PMT images were added, and image adjustments were processed using Photoshop. (**c**–**h**) Area of RH-GFP tachyzoites after treatment, quantified with Imaris per 1.10 × 10^6^ µm^2^. The control data presented here are identical to those shown in (**c–h**), m. EtOH in Ad: ethanol diluted in double-distilled water; EtOH in PBS: ethanol diluted in PBS; H_2_O_2_ in PBS: hydrogen peroxide diluted in PBS; chlorcresol in PBS: p-chloro-m-cresol diluted in PBS; HFAE + PA in PBS: o-hydroxydiphenyl fatty acid eutectic component + peracetic acid diluted in PBS; ns: no significant difference compared to the positive control. All other groups showed significant differences (**** *p* < 0.0001) compared with the positive control. Individual *p*-values are provided in Appendix A. Error bar: standard error of the mean.

**Table 1 microorganisms-13-01517-t001:** Experimental setting.

Reagent	Concentration (%)	Incubation Medium	Incubation Time (m)
Negative control/−	-	PBS	30
Positive control/+ (Paraformaldehyde)	4	PBS	30
Ethanol	70	dH_2_O	30
Ethanol	50	dH_2_O	30
Ethanol	30	dH_2_O	30
Ethanol	10	dH_2_O	30
Ethanol	0	dH_2_O	10
Ethanol	70	PBS	30
Ethanol	50	PBS	30
Ethanol	30	PBS	30
Ethanol	10	PBS	30
Ethanol	70	PBS	10
Ethanol	50	PBS	10
Ethanol	30	PBS	10
Ethanol	10	PBS	10
H_2_O_2_	0.3	PBS	30
H_2_O_2_	30	PBS	10
H_2_O_2_	10	PBS	10
H_2_O_2_	3	PBS	10
H_2_O_2_	0.3	PBS	10
Chlorcresol	4	PBS	30
Chlorcresol	1	PBS	30
Chlorcresol	0.4	PBS	30
Chlorcresol	4	PBS	10
Chlorcresol	1	PBS	10
Chlorcresol	0.4	PBS	10
HFAE + PA	4.5	PBS	30
HFAE + PA	1	PBS	30
HFAE + PA	4.5	PBS	10
HFAE + PA	1	PBS	10
**Agent**	**Temperature (°C)**	**Incubation medium**	**Incubation time (m)**
Heating	60	PBS	30
Heating	50	PBS	30
Heating	40	PBS	30
Heating	60	PBS	10
Heating	50	PBS	10
Heating	40	PBS	10

PBS, Phosphate-buffered saline; dH_2_O, double-distilled water; Chlorcresol, p-chloro-m-cresol; HFAE + PA, o-Hydroxydiphenyl fatty acid eutectic + peracetic acid.

## Data Availability

The original contributions presented in the study are included in the article/Appendix A, further inquiries can be directed to the corresponding author.

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
