# Peer review of "Low Tenacity of Toxoplasma gondii Tachyzoites In Vitro"

_microorganisms, 2025, doi:10.3390/microorganisms13071517_

Round 1

Reviewer 1 Report

Comments and Suggestions for Authors

In the article entitled “Low tenacity of Toxoplasma gondii tachyzoites in vitro Low tenacity of Toxoplasma gondii tachyzoites in vitro”, Grochow and collaborators evaluate the activity of some compounds on the viability of T. gondii tachyzoites. 
The paper is interesting. However, there are some improvements that could be made.

Comment 1. Throughout the manuscript, the words in vitro and in vivo should be italicized.
Comment 2. The authors should explain the origin of the MARC-145 cell line.
Comment 3. The authors should include in the material and methods the origin of all reagents (trademark, country of origin...). Information is only available for some of the reagents in the article.

Author Response

Summary: In the article entitled “Low tenacity of Toxoplasma gondii tachyzoites in vitro Low tenacity of Toxoplasma gondii tachyzoites in vitro”, Grochow and collaborators evaluate the activity of some compounds on the viability of T. gondii tachyzoites. 
The paper is interesting. However, there are some improvements that could be made.

Response : We thank the reviewer for the positive evaluation of our manuscript. All changes have been highlighted in yellow in the revised version.

Comments 1: Throughout the manuscript, the words in vitro and in vivo should be italicized.

Response 1: We appreciate the reviewer’s attention to detail. The formatting has been corrected accordingly throughout the manuscript.

Comments 2: The authors should explain the origin of the MARC-145 cell line.

Response 2: Thank you for this suggestion. We have now included a more detailed description of the origin of the MARC-145 cell line in the manuscript.

Comments 3: The authors should include in the material and methods the origin of all reagents (trademark, country of origin...). Information is only available for some of the reagents in the article.

Response 3: We agree with the reviewer’s comment and have revised the Materials and Methods section to provide consistent and complete information regarding the origin (manufacturer and country) of all reagents used.

Reviewer 2 Report

Comments and Suggestions for Authors

This article presents a study of the efficacy of different inactivation strategies for T. gondii. The article is well structured and the methods used are described in detail, facillitating the replication for other authors. But the topic do not seem very interesting for the readers of Microorganisms, due it seems to be like a extension of comprobation of know strategies of disinfection with several agents. Even, the p-chloro-m-cresol assay is not performed as the manufacturer recommendations, but the contact time was the same as the other agents for comparison. Nevertheless, it can be suitable for publication if editors consider it valuable, due there is not significant problems with the article structure. I csan mention a few minor changes that must be done before publication:

  • In vitro and in vivo must be in kursive.
  • The scale bars must be in the figure. It is ok to mention in the figure description but the units must be in the figure itself.
  • Delete "A" before "The" in line 307.

I recommend minor revison of the text.

Author Response

Summary: This article presents a study of the efficacy of different inactivation strategies for T. gondii. The article is well structured and the methods used are described in detail, facillitating the replication for other authors. But the topic do not seem very interesting for the readers of Microorganisms, due it seems to be like a extension of comprobation of know strategies of disinfection with several agents. Even, the p-chloro-m-cresol assay is not performed as the manufacturer recommendations, but the contact time was the same as the other agents for comparison. Nevertheless, it can be suitable for publication if editors consider it valuable, due there is not significant problems with the article structure. I csan mention a few minor changes that must be done before publication:

Response: We thank the reviewer for the constructive feedback and careful evaluation. All changes have been highlighted in yellow in the revised manuscript.

Comments 1: In vitro and in vivo must be in kursive.

Response 1: Thank you for the suggestion. All occurrences of in vitro and in vivo have been italicized as recommended.

Comments 2: The scale bars must be in the figure. It is ok to mention in the figure description but the units must be in the figure itself.

Response 2: We appreciate this comment and have revised the figures accordingly. Scale bars with appropriate units are now included directly in the figure panels.

Comments 3: Delete "A" before "The" in line 307.

Response 3: Thank you for pointing this out. We have corrected the sentence as suggested.

Comments 4: I recommend minor revison of the text.

Response 4: We thank the reviewer for this helpful recommendation. We have carefully revised the manuscript to improve clarity and consistency throughout.

Reviewer 3 Report

Comments and Suggestions for Authors

The authors compared the effects of various inactivation methods using GFP-containing Toxoplasma gondii tachyzoites. The article is well thought out. But the author needs to improve the presentation of experimental data. A common problem with the article is that the p-values of the experimental groups that were significantly different from the positive control were not shown in the picture.

As well as some of the author's descriptions make me confused:

line 273-275 mentions “indicating significant growth compared to the positive control”, but the corresponding Figure 2d shows the ns.

Line 287-289 mentions “These values indicate strong growth compared to the positive control.” but the corresponding Figure 2f shows the ns.

In addition to this, it is recommended that in the keywords, the author should add the inactivation methods used in this paper

Author Response

Summary: The authors compared the effects of various inactivation methods using GFP-containing Toxoplasma gondii tachyzoites. The article is well thought out. But the author needs to improve the presentation of experimental data. A common problem with the article is that the p-values of the experimental groups that were significantly different from the positive control were not shown in the picture.

Response: We are grateful to the reviewer for this valuable feedback. We have now added the relevant significance levels to the figures to clarify statistical differences. All modifications are highlighted in yellow.

Comments 1: As well as some of the author's descriptions make me confused:

line 273-275 mentions “indicating significant growth compared to the positive control”, but the corresponding Figure 2d shows the ns.

Response 1: Thank you for this important observation. The original figure did not include significance markers for 10%, 10'. We have now updated the figure accordingly, so that statistical differences are clearly indicated.

Comments 2: Line 287-289 mentions “These values indicate strong growth compared to the positive control.” but the corresponding Figure 2f shows the ns.

Response 2: We appreciate the reviewer highlighting this discrepancy. While the difference is not statistically significant, parasite growth was observed, indicating that the treatment is insufficient for complete inactivation. We have revised the text to clarify this point and avoid misinterpretation.

Comments 3: In addition to this, it is recommended that in the keywords, the author should add the inactivation methods used in this paper

Response 3: Thank you for the helpful suggestion. We have added the following keywords to improve searchability and relevance: disinfection, chemical treatment, heat inactivation

Round 2

Reviewer 3 Report

Comments and Suggestions for Authors

The author corrected the errors and answered the questions appropriately.